# On the Irrationality of Being in Two Minds

**DOI:** 10.3390/e22020174

**Published:** 2020-02-04

**Authors:** Shahram Dehdashti, Lauren Fell, Peter Bruza

**Affiliations:** School of Information Systems, Queensland University of Technology, Brisbane 4000, Australia; shahram.dehdashti@qut.edu.au (S.D.); l3.fell@qut.edu.au (L.F.)

**Keywords:** bistable probabilities, human decision making, causal cognition, quantum cognition

## Abstract

This article presents a general framework that allows irrational decision making to be theoretically investigated and simulated. Rationality in human decision making under uncertainty is normatively prescribed by the axioms of probability theory in order to maximize utility. However, substantial literature from psychology and cognitive science shows that human decisions regularly deviate from these axioms. Bistable probabilities are proposed as a principled and straight forward means for modeling (ir)rational decision making, which occurs when a decision maker is in “two minds”. We show that bistable probabilities can be formalized by positive-operator-valued projections in quantum mechanics. We found that (1) irrational decision making necessarily involves a wider spectrum of causal relationships than rational decision making, (2) the accessible information turns out to be greater in irrational decision making when compared to rational decision making, and (3) irrational decision making is quantum-like because it violates the Bell–Wigner polytope.

## 1. Introduction

Understanding how humans form decisions is of great interest in a range of areas, and modeling decision making is important to psychology, social science, politics, economics, computer science, and cognitive science. Current models of human decision making rely on Bayesian probabilities, so much so that the term “Bayesian Cognition” has become mainstream [1,2]. Bayesian models account for rational decision making where “rationality” is defined by the laws of probability theory. However, decades of research have found that a whole range of human judgement deviates substantially from what would be normatively correct according to logic or probability theory.

Theories proposed to account for the deviation of human decision makers from rationality include bounded rationality [3], dual process theories [4], models in quantum cognition [5,6,7,8,9], and a growing list of cognitive biases and heuristics [10]. Dual process theories are prominent and widespread within many fields of psychological science [11]. Whilst there are multiple variations of these theories, all have in common the thesis that two processes are employed in human decision making: One fast, intuitive system sometimes prone to error (System 1), and a second slower, more controlled process based on rational thought (System 2). The extent to which either or both of these are employed is dependent on the relative cognitive resources and time that each typically consumes.

System 1 is fast and requires few cognitive resources and little effort [4], so is often considered the default system [12]. System 1 can be termed “irrational" because the intuition, biases, and heuristics that typify this system often do not adhere to the laws of logic or probability theory. In contrast, System 2 involves controlled analytic thought and is considered to be primarily logical and rational. It requires conscious activation and is a significant drain on cognitive resources [4].

Many of the heuristics employed by System 1 can be considered in terms of *attribute substitution* [10]. Put simply, humans tend to substitute a difficult problem for a more plausible, easier alternative. An example can be seen in the famous illustration of the conjunction fallacy, where participants overestimate the likelihood of Linda to be both a bank teller *and* a feminist, compared to her being a bank teller [13]. Rather than employing System 2 to conduct a rational estimation of probabilities, participants instead substitute the easier representativeness bias to conclude that Linda must be a feminist and that, therefore, the conjunction is the most likely option.

It is easy to see that the assumed rationality of Bayesian Cognition aligns more closely with System 2. The picture from the literature seems to suggest that human decision makers are often irrational because they employ System 1, even when employing System 2 would result in a rational outcome.

Although rational models of human decision making have become prominent and have achieved much success, there has been an emergence of models based on an alternative probabilistic framework drawn from quantum theory [6,7,8,9]. These quantum models show promise in addressing decision making that would normally be considered irrational [14,15]. This article continues this line of research by proposing a model for irrational decision making based on the notion of a bistable probability. The term “bistable” aims to capture the intuition of two sometimes competing systems involved in decision making and consequently the decision maker’s being caught between two “minds”.

The view taken in this paper is that decisions based on intuition, i.e., made by System 1, can sometimes result in a different outcome than judgements based on rational probability, i.e., made by System 2. When this happens, we deem the decision making to be “irrational”, as it deviates from rational judgement. However, the theory presented in the following sections is agnostic to how Systems 1 and 2 interact. It is the deviation from rational judgement that is the core issue.

We will show that bistable probabilities allow (ir)rational decision making to be systematically investigated. Irrationality raises questions such as the following: (1) How does irrationality affect the probabilistic judgement of causality? (2) Does irrationality affect the amount of information available for decision making? (3) How do irrational decisions relate to rational probabilistic judgements? These questions will be addressed in the following sections.

## 2. Bistable Parameters and Bistable Projection Operators

In order to model the deviation of irrational decision making from rational decision making, we propose a deformed probability, introducing what we term a “bistable parameter”. The purpose of this parameter is to capture the disagreement between System 1 and System 2. In doing so, the degree of substitution of a simple heuristic-based inference in place of a knowledge-based, classically Bayesian, inference [16] is also accounted for.

Similarly to the noise model proposed by [17,18], bistable probabilities are founded on an event space featuring two probabilities associated with a given decision outcome. The two probabilities relate to System 1 and System 2. For example, consider the scenario depicted in Figure 1, where System 2 is assumed to mediate the decisions of System 1. There are two decision outcomes *A* and *B*. System 1 opts for outcome *A* with probability *k*. System 2 may intervene and alter the response of System 1 with probability 1−p, resulting in choice *B*, or agree with System 1 and remain with choice *A* with probability *p*. The final probability for choosing option *A* is a function of both *k* and *p*: $P_{k}\left(A\right)=1-p-k+2kp$. Note that when k=1, System 2 fully determines the final judgement, which is hence considered “rational”. When 0≤k<1, the probability of the final judgement is deformed by a degree of irrationality. When k=0, the final outcome is considered “irrational" because it is the converse of the rational judgement, i.e., $P_{k=1}\left(A\right)=p$ vs. $P_{k=0}\left(A\right)=1-p$. When $k=0.5,P_{k}\left(A\right)=0.5$ irrespective of System 2’s mediation. This reflects the situation when the decision maker is caught between two minds with no means of resolving the conflict between the two, so the ultimate decision is random. Finally, System 2 “agrees" with System 1 when $P_{k}\left(A\right)=k$. This occurs when p=1, which implies that no mediation is being employed by System 2.

Relating this to an example commonly used in the literature [13], we consider the following question: Is a person, Linda, more likely to be (a) a bank teller, or (b) a bank teller and active in the feminist movement? Without any additional information about Linda (for now, we will consider the question in the absence of the usually accompanying passage of background text), one might assume that the option that is logically more likely is option a), and one might assign a higher probability to this option. The reason that this is the logical and rational choice is tied to the rule that the probability of a conjunction of two propositions cannot be higher than the probability of either of the individual propositions (P(A∧B)≤P(A)). One’s intuitive response to this question might be to choose the simpler option, a), illustrated as a high value for *k*, for example, 0.8. The parameter *p* determines the degree to which System 2 allows the intuition of System 1 to determine the final decision. Due to the fact that the option deemed more probable by System 1 would also be favored by a rationally driven System 2, we assign a high value for *p*, for example, 0.9. Taking these values of *k* and *p* and following our model, we find a probability of 0.74 for the final decision of option a), that Linda is a bank teller, to be more likely.

Let us now consider the more complete example of the Linda problem, where a passage of text is presented prior to the above question. This text provides additional background information about Linda that tends to accord with a representation of someone who is active in the feminist movement. The question, coupled with the presentation of the background information, provides a classic demonstration of the conjunction fallacy, where the conjunction of two propositions (option b) is judged to be more probable than one of the propositions (option a). This has been attributed to a judgement heuristic labelled representativeness [13], to which System 1 is prone [10]. In our model, we reflect the likely choice of a System-1-driven judgement by assigning a low value for *k*, for example, 0.3. To determine a value of *p*, we can assign a low value based on the premise that the rational thought process employed by System 2 is unlikely to allow the erroneous intuition of System 1 to determine the final decision, and we might assign a value of 0.2, for example. Using these values for *k* and *p* in our model, we find a probability of 0.62 for the final decision of option (a), that Linda is a bank teller, to be more likely.

However, due to the fact that System 2 is constrained in terms of time and cognitive resources [4], the probability that System 2 will differ from or mediate the intuition of System 1 depends on a range of factors, including time constraints, motivation, and availability of cognitive resources. If one or a combination of these factors is present in such a way that it would be reasonable to assume that System 2 is less likely to intervene, we can instead choose a higher value of *p*, for example, 0.7. Using the same value for *k*, 0.3, we find that it is now option (b), that Linda is a bank teller and active in the feminist movement, that has the higher probability in the final decision (0.62), which is illustrative of the conjunction fallacy.

It should be noted that, although the preceding example frames the model in terms of a process (i.e., System 1 providing an intuition for which System 2 may or may not intervene and override), this model is not a dynamic one. It is agnostic to the order of System 1 and 2 and incorporates a time element only in the selection of a value of *p* (i.e., in determining the opportunity that System 2 might have to intervene in the above example). For a dynamic model, see [19]. We do not claim that defining irrationality in the preceding way presents a complete picture. In fact, irrationality can be considered in different ways, e.g., in relation to ideas and theories about anti-realism [20] or to some theories about biases and perceptions in cognitive science [17,18,21,22,23]. However, by parameterizing irrationality using a bistable parameter, irrationality can be investigated in a systematic way.

Whilst comparisons have been made between a similar noise model [24] and models based on quantum formalism [25], we show that the bistability model developed in the present paper can be encapsulated within a quantum framework. Bistable probabilities can be expressed as normalized positive-operator-valued (POV) measures [26]. A projection in quantum mechanics is defined by using orthogonal states, $|\phi_{i}\rangle$, i.e.,
(1)$$\pi_{i}=|\phi_{i}\rangle \langle \phi_{i}|,$$
with the following conditions:(2)$$\pi_{i}\pi_{j}=\delta_{i,j}\pi_{i},\;\mathrm{and}\;\sum_{i=1}^{d}\pi_{i}=I,$$
in which $\delta_{i,j}$ is the Kronecker delta and *d* is the dimension of the Hilbert space. However, a general measurement in quantum mechanics is described by means of a POV projection acting on the quantum state defined in the complex Hilbert space [27]. Despite the fact that orthogonality is not a necessary condition with respect to such projections, which means that the results of two measurements following each other are not the same, i.e., $E_{i}E_{i}\neq E_{i}$, the second condition holds:(3)$$\begin{array}{c} \sum_{i=1}^{d}E_{i}=I. \end{array}$$

In a binary system, a set of unsharp projections is defined as follows [28],
(4)$$\begin{array}{c} E_{\pm }^{n}=\frac{1}{2}I_{2\times 2}\pm \frac{\eta }{2}\sigma \cdot \hat{n},\;0\leq \eta \leq 1, \end{array}$$
in which $I_{2\times 2}$ is the two-dimensional identity matrix, η∈[0,1] is the so-called noisy parameter, σs are the standard Pauli matrices,
(5)$$\begin{array}{c} \sigma_{x}=\left(\begin{array}{cc} 0 & 1\\ 1 & 0 \end{array}\right),\sigma_{y}=\left(\begin{array}{cc} 0 & -i\\ i & 0 \end{array}\right),\sigma_{z}=\left(\begin{array}{cc} 1 & 0\\ 0 & -1 \end{array}\right), \end{array}$$
and $\hat{n}=\left(\sin\vartheta \cos\varphi ,\sin\vartheta \sin\varphi ,\cos\vartheta \right)$ gives the direction of projections in the Bloch sphere.

Although in quantum cognition, the noise parameter η is attributed to memorylessness and weak interaction [29], we ascribe it to the impact of irrationality on decision making.

We consider a linear map η=2k−1 together with an extended noise interval η∈[−1,1]. Hence, we obtain the bistable projection as follows:(6)$$\begin{array}{c} P_{\pm }^{\hat{n}}=\left(1-k\right)I_{2\times 2}\pm \left(2k-1\right)\pi_{\pm }^{\hat{n}} \end{array}$$
in which *k* is a bistable parameter and defined in the interval 0≤k≤1. $\pi_{\pm }^{\hat{n}}$ is a positive value projection and is defined by
(7)$$\begin{array}{ccc} \pi_{\pm }^{\hat{n}} & = & \frac{1}{2}\left[I_{2\times 2}\pm \sigma \cdot \hat{n}\right]\\ & = & \left(\begin{array}{cc} \cos^{2}\frac{\vartheta }{2} & e^{-i\varphi }\sin\frac{\vartheta }{2}\cos\frac{\vartheta }{2}\\ e^{i\varphi }\sin\frac{\vartheta }{2}\cos\frac{\vartheta }{2} & \sin^{2}\frac{\vartheta }{2}. \end{array}\right) \end{array}$$

By using an analogy with quantum mechanics, we postulate that the probability of an irrational decision is given by the expectation value of the associated bistable projection, that is,
(8)$$P_{\pm }\left(k,n\right)=\langle \psi |P_{\pm }^{\hat{n}}|\psi \rangle ,$$
in which the bistable projection $P_{\pm }^{\hat{n}}$ is defined by the Equation (Equation 6) and $|\psi \rangle ={\left(\sqrt{p},\sqrt{1-p}\right)}^{T}$ gives the probability of a rational decision. Note that in the special case in which the bistable projection (Equation 8) is defined in the direction *z*,
(9)$$\begin{array}{c} P_{+}^{z}=\left(\begin{array}{cc} k & 0\\ 0 & 1-k \end{array}\right),\;P_{-}^{z}=\left(\begin{array}{cc} 1-k & 0\\ 0 & k \end{array}\right). \end{array}$$

We can reproduce the output of bistable decision making, i.e.,
(10)$$\begin{array}{ccc} P_{k}\left(+\right) & = & \left(\sqrt{p}\;\sqrt{1-p}\right)\left(\begin{array}{cc} k & 0\\ 0 & 1-k \end{array}\right)\left(\begin{array}{c} \sqrt{p}\\ \sqrt{1-p} \end{array}\right)\\ & = & 1-p-k+2kp \end{array}$$
(11)$$\begin{array}{ccc} P_{k}\left(-\right) & = & \left(\sqrt{p}\;\sqrt{1-p}\right)\left(\begin{array}{cc} 1-k & 0\\ 0 & k \end{array}\right)\left(\begin{array}{c} \sqrt{p}\\ \sqrt{1-p} \end{array}\right)\\ & = & p+k-2kp. \end{array}$$

## 3. Causality

In this section, we address the question of what irrationality means for the probabilistic judgement of causality.

### 3.1. Inferring Causality

Let us start with Reichenbach’s principle: We assume that two variables *Y* and *Z* are found to be statistically dependent; then, (*i*) either *Y* is part of a cause of *Z* or *Z* is part of a cause of *Y*, as shown in the plot Figure 2a, and (*ii*) *Y* and *Z* have a common cause *X*, illustrated in the plot Figure 2b. Consequently, causal independence implies statistical independence, i.e., P(Y,Z)=P(Y)P(Z) and P(Y,Z|X)=P(Y|X)P(Z|X), in which *X* is the collection of all variables acting as common causes [30]. Therefore, causality can determined based on probabilities. In fact, the following relation:(12)$$\begin{array}{c} P_{k}\left(Y=i\right)P_{k}\left(Z=j\right)\neq P_{k}\left(Y=i,Z=j\right) \end{array}$$
in which i,j can be any observable outcomes ±, characterizes a causal relationship, i.e., either *Y* causes *Z* or *Z* causes *Y*. Conversely, probabilities that are not factorizable are non-causal, and are therefore un-deterministic [30]. For example, by using (Equation 10), and assuming a rational non-causal relationship between two variables (k=1), Y=+ and Z=+; i.e.,
(13)$$P_{k=1}\left(Y=\pm ,Z=\pm \right)=P_{k=1}\left(Y=\pm \right)P_{k=1}\left(Z=\pm \right).$$

Then, by considering (0≤k<1), we have
(14)$$\begin{array}{c} 1-P(Y=+,Z=+)-k+2kP(Y=+,Z=+)\neq \\ \left[1-P(Y=+)-k+2kP(Y=+)\right]\left[1-P(Z=+)-k+2kP(Z=+)\right]. \end{array}$$

In other words, the presence of irrationality eliminates the cognitive agent’s ability to recognize independence, and potentially spurious causality is discerned by the agent. The tendency to overestimate relationships between events is seen in many heuristics stemming from System 1 processes, such as the illusion of validity bias [31], spontaneous causal inference [32], and illusions of causation [33]. In addition, this effect is exemplified in classically irrational thought processes, such as superstition and Obsessive Compulsive Disorder, where actions are erroneously causally linked to positive or negative events in an individual’s mind. In addition, a comprehensive empirical study of human causal reasoning found that participants committed violations of the Markov condition, which prescribes when variables are independent of each other [34]. For example, in a common cause network (Figure 2b), the Markov condition entails that variables *Y* and *Z* are conditionally independent, i.e., non-causally related, when the value of *X* is known. However, participants deemed *Y* and *Z* to influence each other when they were supposedly independent because of the Markov condition. These violations were present in experimental conditions which were specifically designed to distinguish between the processing of System 1 and System 2. These findings suggest that the distortion is not always due to System 1.

The bistable model can also account for situations where the converse occurs; namely, a rational causal relation (see Equation (Equation 12)) is distorted into a non-causal relation. Consider a common effect network where X→Z and Y→Z, (Figure 2a). If the value of *Z* is known, then *X* and *Y* become conditionally dependent. However, they may be irrationally deemed to be conditionally independent, and hence not causally related—for example, Ozone → Humidity and Air Pressure → Humidity. If it becomes known that the Humidity is high, then, rationally, Ozone and Air pressure become conditionally dependent. However, irrationally, one might see these as independent. The distortion also corresponds to violations of the “causal faithfulness condition" which states that variables that are causally connected are probabilistically dependent [35].

### 3.2. Causal Strength Criterion

By employing causal strength and its power, we study the impact of irrationality on a final judgement. The definition of the causal strength measure given by [36,37,38,39],
(15)$$\begin{array}{c} \Delta P=P(Y|X)-P(Y|\neg X), \end{array}$$
is independent of P(X), but we should note that the causal strength is low if P(Y|¬X) is high. Therefore, as another criterion, the power of causal strength κ is suggested:(16)$$\begin{array}{c} \kappa =\frac{\Delta P}{P(\neg Y|\neg X)}, \end{array}$$
in which ΔP is given by relation (Equation 15).

By using relations (Equation 15) and (Equation 16), for a final judgement with probability $P(X_{out}=+)=1-p-k+2kp$, the causal strength measure and its power are obtained as
(17)$$\begin{array}{ccc} \Delta P_{k} & = & P\left(X_{out}=+|k=+\right)-P\left(X_{out}=+|k=-\right)=k+p-1\\ \kappa & = & \frac{\Delta P_{k}}{P(X_{out}=-|p=-)}=\frac{k+p-1}{\left(1-p)(1-k\right)}, \end{array}$$
in which the causal strength measure and its power clearly depend on the probability *k*. In fact, the relations of (Equation 17) indicate that increasing irrationality, i.e., lower levels of *k*, decreases the causal strength and its power.

We now consider a situation in which an outcome of variable *X* causes the outcome of variable *Y*. When the level of bistability changes, what effect does it have on the cause–effect relationship between $X_{out}$ and $Y_{out}$? In other words, if we rationally assume that there is a cause–effect relationship between two variables, what can we say about the cause–effect relationship between the final outcomes? For simplicity, we assume that the bistable projections of variable *X* are given by the relations in (Equation 9) and the associated bistable projections of variable Y=± are considered in the *x*-axis direction:(18)$$\begin{array}{c} P_{k}^{Y}\left(+\right)=\left(\begin{array}{cc} \frac{1}{2} & k-\frac{1}{2}\\ k-\frac{1}{2} & \frac{1}{2} \end{array}\right),P_{k}^{Y}\left(-\right)=\left(\begin{array}{cc} \frac{1}{2} & \frac{1}{2}-k\\ \frac{1}{2}-k & \frac{1}{2} \end{array}\right). \end{array}$$

Hence, the causal strength measure is given by
(19)$$\begin{array}{ccc} \Delta P_{k} & = & P_{k}\left(Y=+|X=+\right)-P_{k}\left(Y=+|X=-\right)\\ & = & \langle P_{k}^{Y}\left(+\right)P_{k}^{X}\left(+\right)\rangle -\langle P_{k}^{Y}\left(+\right)P_{k}^{X}\left(-\right)\rangle \\ & = & \frac{1}{2}\left(1-k-p+2kp\right) \end{array}$$
(20)$$\begin{array}{ccc} \kappa & = & \frac{1-p-k+2kp}{p+k-2kp+\left(1-2k\right)\sqrt{p(1-p)}} \end{array}$$
in which we assume that a real Hilbert space describes the original probability of the system, i.e., $|\psi \rangle ={\left(\sqrt{p},\sqrt{1-p}\right)}^{T}$. Again, the above-mentioned relations (Equation 19) and (Equation 20) illustrate that the causal strength decreases by increasing the role of the bistable parameter, that is, decreasing the bistable parameter. In addition, we note that if the value of the bistable parameter is equal to k=0.5, both criteria approach zero, which means that it is not possible to establish a cause–effect relationship between variables $X_{out}$ and $Y_{out}$.

## 4. Polytopes of Bistable Probabilities

We now consider a joint decision scenario where two decisions $E_{1}$ and $E_{2}$ with probability $P_{1}$ and $P_{2}$, with $P_{12}$ denoting the joint probability. There are necessary and sufficient conditions for the rational values of $P_{1}$, $P_{2}$, and $P_{12}$, as in the following [40]:(21)$$\begin{array}{cc} & 0\leq P_{i}\leq 1,\;P_{i}\geq P_{12},\;P_{12}\geq 0,\;P_{1}+P_{2}-P_{12}\geq 1, \end{array}$$
where i=1,2. The above-mentioned relations are so-called Boole’s conditions [40]. By considering a three-dimensional space in which $P_{1}$, $P_{2}$, and $P_{12}$ are coordinate axes, Boole’s conditions (Equation 21) construct a polytope. Therefore, each point that fulfills Boole’s conditions is a potential rational choice. The plot in Figure 3a illustrates a geometrical representation of this polytope. Now, based on this interpretation, by which the polytope indicates the volume of information that can be accessed, we consider two bistable outputs and their conjunction operator. To obtain the polytope structure, we consider the geometrical structure of the truth table of bistable outputs, which is the same as the truth table of non-bistable outputs. We can obtain a collection of linear inequalities for probabilities of bistable outputs:(22)$$\begin{array}{ccc} \left(P_{1}^{k},P_{2}^{k},P_{12}^{k}\right) & = & \lambda_{1}\left(0,0,0\right)+\lambda_{2}\left(0,1,0\right)\\ & + & \lambda_{3}\left(1,0,0\right)+\lambda_{4}\left(1,1,1\right) \end{array}$$
in which $\lambda_{i}\geq 0$, for i=1,⋯,4 and $\sum_{i=1}^{4}\lambda_{i}=1$. The following polytope’s equations describe information which is accessible:(23)$$\begin{array}{c} \left(2k-1\right)P_{i}-P_{12}^{k}\geq k-1,\;i=1,2, \end{array}$$
and
(24)$$\begin{array}{c} \left(2k-1\right)\left(P_{1}+P_{2}\right)-P_{12}^{k}\leq 2k-1, \end{array}$$
while we keep in mind the following classical identity:(25)$$\begin{array}{c} P\left(E_{1}\vee E_{2}\right)=P\left(E_{1}\right)+P\left(E_{2}\right)-P\left(E_{1}\wedge E_{2}\right). \end{array}$$

In addition, in the case where k=1/2, $P_{12}^{k}$ is independent of probabilities $P_{i},i=1,2$, $0\leq P_{12}^{k}\leq 1/2$. Plots (a)-(f) in Figure 3 illustrate polytopes of the outputs for different values of *k*, that is, k=1,0.9,⋯,0.5. This figure indicates that the accessible information increases as the bistable parameter decreases. In other words, increasing irrationality in decision making results in an increase in the amount of information accessible to the cognitive agent transacting the decision.

We define a new concept, *“pure irrational information volume”* (PIIV), as the difference in volumes of a polytope of irrational decision making (0≤k<1) compared to the polytope of rational decision making (k=1), that is, Δ(k)=V(k)−V(k=1). This indicates the extra information that is accessible in irrational decision making. In the case of bistablity, the PIIV is given by:(26)$$\begin{array}{c} \Delta \left(k\right)=\frac{1-k}{6}+\frac{1-k}{2}\sqrt{\frac{2{\left(2k-1\right)}^{2}+1}{{\left(2k-1\right)}^{2}+1}}. \end{array}$$

In fact, PIIV Δ(k) is a criterion by which we draw a comparison between the amount of information accessible by System 1 and System 2. Figure 4 indicates function Δ(k) as a function of *k*. The plot illustrates that decreasing parameter *k*, i.e., increasing irrationality, causes the accessible information to increase. This increased availability of information to the irrational decision maker could be interpreted in terms of *exploration* [41], where irrationality may be seen to co-occur with the search for novel information or with increased actions or strategies available to the cognitive agent. To place this into the context of attribute substitution, the PIIV accounts for the wider variety of available heuristics employed by a decision maker who is relying on System 1.

## 5. The Bell–Wigner Polytope of Irrational Decision Making

When studying probability theory at school, toy examples such as tossing coins or pulling colored marbles from a bag are often used. When a red marble is drawn from a bag, it is unquestionably assumed that it already had the property of being red before it was pulled out of the bag. Its property of pre-existing redness is simply noted when the marble is retrieved, thereby contributing to the relative frequency of red marbles sampled from the bag. When George Boole was developing probability theory in the mid-1850s, he did so by considering what he called the “conditions of possible experience”. He formalized his intuitions into inequalities that the relative frequencies must satisfy. For example, for events $E_{1}$ and $E_{2}$ with relative frequencies p−1 and $p_{2}$ and where $p_{12}$ denotes the frequency of the joint event $E_{1}\wedge E_{2}$,
(27)$$\begin{array}{c} 0\leq p_{i}\leq 1,\;i=1,2,3. \end{array}$$
(28)$$\begin{array}{c} 0\leq p_{ij}\leq \min\left\{p_{i},p_{j}\right\},\;i,j=1,2,3. \end{array}$$
(29)$$\begin{array}{c} p_{i}+p_{j}-p_{ij}\leq 1,\;i,j=1,2,3. \end{array}$$
(30)$$\begin{array}{c} p_{1}+p_{2}+p_{3}-p_{12}-p_{13}-p_{23}\leq 1.\; \end{array}$$
(31)$$\begin{array}{c} p_{1}-p_{12}-p_{13}+p_{23}\geq 0\; \end{array}$$
(32)$$\begin{array}{c} p_{2}-p_{12}-p_{23}+p_{13}\geq 0\; \end{array}$$
(33)$$\begin{array}{c} p_{3}-p_{13}-p_{23}+p_{12}\geq 0\;. \end{array}$$

Pitowsky [42] uses the preceding inequalities to define the “Bell–Wigner" polytope. Basically, it is the region within the polytope that defines Boole’s conditions of possible experience. Pitowsky [40] shows that quantum systems do not always adhere to these conditions, meaning that “quantumness" can be identified by regions that are outside of the Bell–Wigner polytope [43]. We will use this property in the following to examine whether irrational decision making conforms to Boole’s condition of possible experience or is quantum-like. For this purpose, three bistable parameters $k_{1},k_{2}$, and $k_{3}$ are used to respectively attenuate the probabilities $p_{1},p_{2}$, and $p_{3}$. These parameters were systematically manipulated and the inequalities were tested for violation. Figure 5 depicts six plots for different values of $k_{3}$, while each plot indicates values of inequality (31) with respect to different values of $k_{1}$ and $k_{2}$. These plots illustrate that the maximum violation of the Bell–Wigner polytope happens whenever just one probability becomes irrational. In other words, decision making is necessarily quantum-like in the presence of irrationality. A future direction is to examine the hypothesized connection between quantum-like decision making and irrationality by using the QTEST framework [44]. For example, QTEST could be used to estimate how far simulated data fit a Bayesian model. A lack of good fit of a Bayesian model could suggest the presence of a quantum-like model. This is because Bayesian models derive from standard probability theory and must therefore be bounded by the Bell–Wigner polytope.

## 6. Conclusions

In this paper, we introduced and studied the mathematical consequences of a bistable probabilistic model which enables degrees of irrationality (that is, disagreement between System 1 and System 2) to be systematically investigated. By means of POV projections, it was shown that the bistable model can be considered part of an overarching quantum formalism. We discussed the implications of the bistable model in terms of the propensity of cognitive agents to spuriously infer causality, the impact of irrationality on causal power, and formalizing the amount of extra information available to the irrational decision maker. Finally, we simulated decision making and demonstrated violations of the Bell–Wigner polytope. Such violations suggest that irrational decision making is quantum-like.

## Figures and Tables

**Figure 1 entropy-22-00174-f001:**
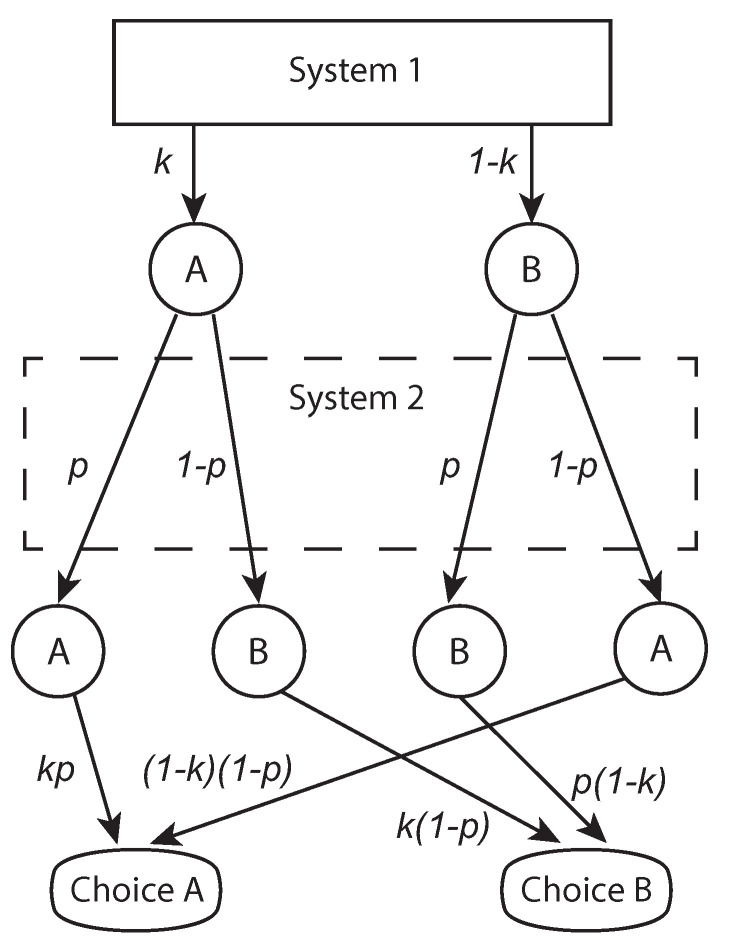
A schematic setup for a bistable model structure.

**Figure 2 entropy-22-00174-f002:**
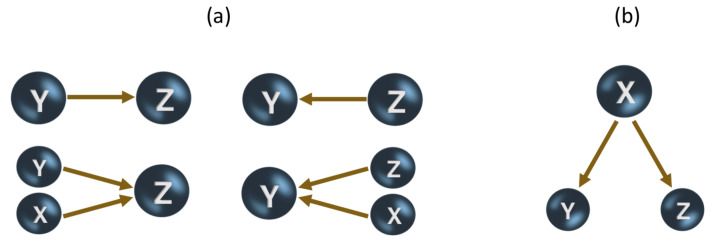
Alternative causal models based on Reichenbach’s principle.

**Figure 3 entropy-22-00174-f003:**
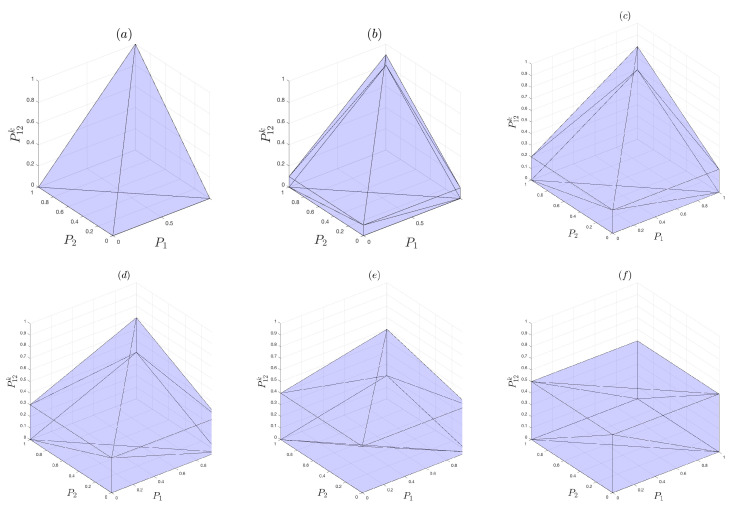
Polytopes for different values of k∈{1.0,0.9,0.8,0.7,0.6,0.5} are respectively shown in plots (**a**)–(**f**).

**Figure 4 entropy-22-00174-f004:**
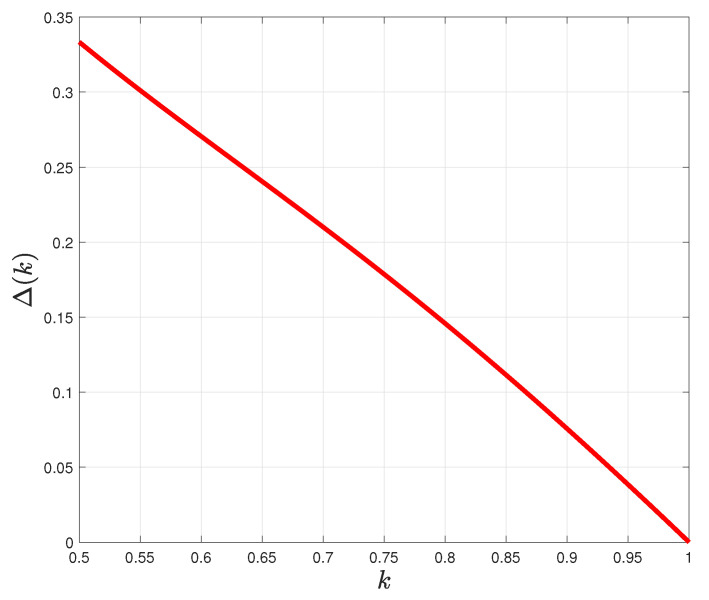
Pure irrational information volume (PIIV), i.e., Δ(k) as a function of the bistable parameter *k*.

**Figure 5 entropy-22-00174-f005:**
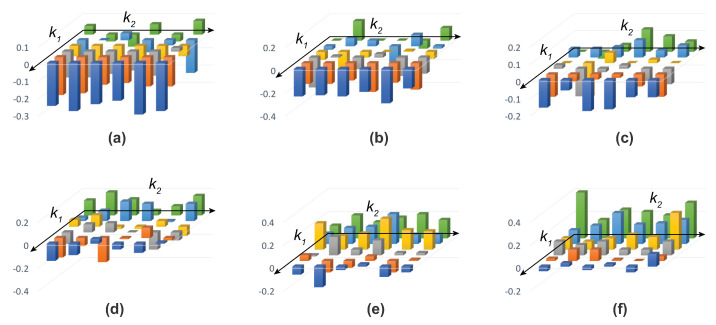
Bell–Wigner polytopes for different values of $k_{3}=1,0.9,\cdots ,0.5$ are illustrated in plots (**a**), (**b**), ⋯, (**f**). Colors differentiate different values of $k_{1}$, with blue signifying $k_{1}=0.5$ and green signifying $k_{1}=1$. Paramater $k_{2}$ varies from 0.5 to 1. Bars below the $k_{1},k_{2}$ plane signify negative probabilities.

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
