# Peer review of "On the Irrationality of Being in Two Minds"

_entropy, 2020, doi:10.3390/e22020174_

Round 1
Reviewer 1 Report
The authors propose and investigate a physical model for irrational decision making. They consider first a classical and then a quantum model based on projection operator. I found the paper well written and presentation clear. I only had some problem when inspecting eq. (3), because no general definition of the quantity E_i is given. It is defined for a binary system in eq. (4), but I was still unable to understand what does the E_\pm^n meand. Also the superscript `n' seem confusing, because the same symbol in eq. (3) denotes the upper bound of the summation. So I guess that `n' should have a `hat', as in eq. (6). Anyway, the E_\pm^n does not appear anywhere later in the text. it would be useful to provide some explanation of the symbol E, and not only provide the reference [26]. Otherwise, I recommend the paper for publication.
Author Response
Re clarification of E_i.
the dimensionality of the Hilbert space has now been specified as $d$ in equation (2) (see blue text line 120 on p4), which prevents confusion with $n$ (as it was previously).
That E_i is not idempotent has been added as another formal characteristic in blue on line 120.
E^n is now used in equation (4) which is clearer than the $k$ used previoulsy.
In addition, more clarification of the noise parameter $\neta$ was added with a supporting reference [29] (line 124 - 126, in blue)
Reviewer 2 Report
In this paper, the authors develop a bistable model of the dual-system view of decision-making and irrationality (System 1 / System 2). They outline how different degrees of irrationality, when System 1 and 2 come into conflict with one another, may lead to violations of the rules of classical probability, characterized by responses that fall outside the Bell-Wigner polytope formed by the systems of inequalities derived from the classical formalism.
In general, the paper was relatively easy to read for the complexity of the subject matter. I could use a little bit more intuition and explanation of some of the topics covered in the paper (outlined below) as well as a bit more content overall to illustrate why this is a potentially interesting approach. Ideally, this might include a re-analysis of some data where people have been shown to violate classical axioms using the newly-developed modeling approach. Otherwise, it can be hard for the reader to really understand how it can be applied and why it would give us new insights about the cognitive processes underlying “irrational” behavior.
Comments:
(p.2, lines 66-68) I could use an explanation of where the formula for P_K(E) is coming from earlier on in the text. Is p the probability of P(E) here? At first, it’s a bit unclear that the formula follows the paths shown in Figure 1, so it could help to reference the path diagrams directly as well. It’s not explicitly stated until Equation 10, and could use a little bit more of an intuitive explanation even there.
Also, k = 0 can still result in different levels of disagreement. If p = .5 then there is no conflict between them, which seems to imply that the extremity of a System 1 belief p is necessarily inversely proportional to the extremity of a System 2 belief when k = 0. I’m not sure I understand the reasoning here – isn’t it possible to have systems in conflict where one system is unsure of a response (e.g., System 1, p = .5) while the other is certain (e.g., System 2, p = 1). Put another way, it seems like it should mechanistically be possible to generate a response from an “irrational” system (System 1) that is nonetheless correct, and so I am a bit confused by diagrams like Figure 1D where irrationality (k = 0) forces a person to make a response that is at odds with the products of System 2. It’s briefly mentioned in the discussion, but many heuristics researchers (Gigerenzer, Hertwig, and colleagues) would probably argue that many “irrational” shortcuts are use because they usually lead us to the correct answer (the same as System 2), rather than in spite of leading us to the incorrect answer.
It’s probably worth mentioning the work of Regenwetter, Davis-Stober, and colleagues (e.g. Transitivity of preferences, 2011; or QTEST papers), who have developed methods for testing models that can be defined as polytopes based on systems of inequalities like the ones used here (Equations 20-32), if only as a future direction. Along these lines, it is potentially an open question as to exactly how far outside the Bell-Wigner polytope ordinary judgments might fall, and this approach could be leveraged to perform tests on empirical data (which would be nice to see, if only to illustrate the usefulness of this approach). The QTEST framework would make this relatively straightforward and I think testing a set of empirical data in this way would be a valuable addition to the manuscript.
Related to this, it is important to be careful about what behaviors or choices are deemed irrational, and a great deal of past work on dual-system approaches leading to irrationality has been rather haphazard from a statistical standpoint. The work of Costello & Watts (2014) illustrated that noise and error propagation can essentially lead to apparent violations of classical probability rules (such as conjunction errors), which means that it is critically important to have an estimate of the certainty of a violation. Using the QTest approach will give classical and Bayesian estimates of how likely the data are (or how likely the hypothesis defined by the polytope is) for these types of problems.
I should note that there is a great deal of work on the dual-system approach and as such the material presented in the paper covers only some dimensions of the distinction between System 1 and System 2. Perhaps most importantly, it doesn’t characterize them in terms of dynamic processes – generally speaking, System 1 is thought to be the one that acts first / most quickly and that System 2 acts second / more slowly. In that sense the path diagram (Figure 1) should almost go in the opposite order (System 1 -> System 2 -> choice), or have System 2 as a moderator of the link between System 1 and the choice / action. Are there any other ways that this approach might be able to incorporate time as an element of the processes leading to choice? One option might be to connect it to the work of Trueblood & Diederich (2018), who implemented the dual process approach in a dynamic diffusion model.
Author Response
We have addressed the comment "Otherwise, it can be hard for the reader to really understand how it can be applied and why it would give us new insights about the cognitive processes underlying “irrational” behavior" in the following way.
We haven’t re-analysed some data, but we have attempted to better ground the theory in terms of a substantial empirical study in human causal reasoning (Rehder 2014). The grounding was implemented in two ways (bottom half of page 6, purple):
The bistable model predicts that rational conditional independence (Markov condition) can be irrationally construed as causality. New content was added to show how this relates to findings where participants regularly violated conditional independence. This added discussion was also placed in the context of System 1 and System 2. New content was added around the converse consequence of our theory: rational causal relationships are irrationally construed as conditional independence. This finding was related to violations of the "causal faithfulness condition"
(p.2, lines 66-68) I could use an explanation of where the formula for P_K(E) is coming from earlier on in the text. Is p the probability of P(E) here?
We have provided a more detailed clarification (page 2, lines 70-82, in purple) that corresponds to a new Figure 1 (which addresses other issues described below)
Also, k = 0 can still result in different levels of disagreement. If p = .5 then there is no conflict between them, which seems to imply that the extremity of a System 1 belief p is necessarily inversely proportional to the extremity of a System 2 belief when k = 0. I’m not sure I understand the reasoning here – isn’t it possible to have systems in conflict where one system is unsure of a response (e.g., System 1, p = .5) while the other is certain (e.g., System 2, p = 1). Put another way, it seems like it should mechanistically be possible to generate a response from an “irrational” system (System 1) that is nonetheless correct, and so I am a bit confused by diagrams like Figure 1D where irrationality (k = 0) forces a person to make a response that is at odds with the products of System 2. It’s briefly mentioned in the discussion, but many heuristics researchers (Gigerenzer, Hertwig, and colleagues) would probably argue that many “irrational” shortcuts are use because they usually lead us to the correct answer (the same as System 2), rather than in spite of leading us to the incorrect answer.
We have expanded on our explanation of this, whilst also providing a new Figure 1 to reflect a common interpretation of how System 1 and System 2 interact. We have attempted to make clearer the fact that one probability denotes the choice System 1 would make (k, in our new explanation), whereas the other denotes the degree to which this choice is deformed or mediated by a System 2 (p). We have also replaced the examples given in the previous version of Figure 1 (B, C & D) with an in-text example showing which values of p and k can demonstrate a situation where System 1 leads to a correct answer (line 81-82, page 2). Whilst we re-arranged the order of System 1 & 2 illustrated in our figure, we note that our model is agnostic as to the order of System 1 and 2 and is not intended as a dynamic model.
It’s probably worth mentioning the work of Regenwetter, Davis-Stober, and colleagues (e.g. Transitivity of preferences, 2011; or QTEST papers), who have developed methods for testing models that can be defined as polytopes based on systems of inequalities like the ones used here (Equations 20-32), if only as a future direction. Along these lines, it is potentially an open question as to exactly how far outside the Bell-Wigner polytope ordinary judgments might fall, and this approach could be leveraged to perform tests on empirical data (which would be nice to see, if only to illustrate the usefulness of this approach). The QTEST framework would make this relatively straightforward and I think testing a set of empirical data in this way would be a valuable addition to the manuscript.
Related to this, it is important to be careful about what behaviors or choices are deemed irrational, and a great deal of past work on dual-system approaches leading to irrationality has been rather haphazard from a statistical standpoint. The work of Costello & Watts (2014) illustrated that noise and error propagation can essentially lead to apparent violations of classical probability rules (such as conjunction errors), which means that it is critically important to have an estimate of the certainty of a violation. Using the QTest approach will give classical and Bayesian estimates of how likely the data are (or how likely the hypothesis defined by the polytope is) for these types of problems.
This is a good suggestion that we have noted as a future direction to be pursued (See bottom of page 10, line 260-266, in purple)
Related to this, it is important to be careful about what behaviors or choices are deemed irrational, and a great deal of past work on dual-system approaches leading to irrationality has been rather haphazard from a statistical standpoint. The work of Costello & Watts (2014) illustrated that noise and error propagation can essentially lead to apparent violations of classical probability rules (such as conjunction errors), which means that it is critically important to have an estimate of the certainty of a violation. Using the QTest approach will give classical and Bayesian estimates of how likely the data are (or how likely the hypothesis defined by the polytope is) for these types of problems.
Whilst time is not specifically dealt with in this model, it can be considered to be part of the process of identifying a value for p. We have expanded an explanation of the model with an example that discusses the selection of p to be based both on the likelihood that System 2 would agree with System 1, and the likelihood that System 2 would be employed – something that may depend on time, motivational factors, availability of cognitive resources, etc. We now refer readers to the dynamic dual process model proposed by Diederich & Trueblood (2018) for a dynamic model. (Top of page 4, second paragraph, in purple)
Round 2
Reviewer 1 Report
The author did not explicitly explain the meaning of the symbol E, but from the inserted explanation I guess that the E_i denotes the result of a measurement. If I am wrong, then it is up to the author to insert some more explanation, but I do not need to see the paper again. In any case, I recomend publication of the manusrcipt.
Reviewer 2 Report
In the revision, the authors have made several changes to address issues I raised in the initial manuscript. In particular, they have clarified the structure of the model (Figure 1 is now substantially clearer), added some examples to ground the theory, and made stronger connections to previous work on dual-systems theory and causal reasoning. I still would have liked an application to empirical data to help demonstrate the usefulness of the approach and how it can be applied to real data, but for a more theoretically-oriented journal (and one the emphasizes more the information theoretic components of the model than the psychological ones) it may be acceptable without this addition.